# Synergistic information supports modality integration and flexible learning in neural networks solving multiple tasks

**Alexandra M. Proca**[1]*, **Fernando E. Rosas**[2,3,4,5], **Andrea I. Luppi**[6,7,8], **Daniel Bor**[9,10], **Matthew Crosby**[1‡], **Pedro A. M. Mediano**[1,9‡]

**1** Department of Computing, Imperial College London, London, United Kingdom, **2** Department of Informatics, University of Sussex, Brighton, United Kingdom, **3** Sussex Centre for Consciousness Science and Sussex AI, University of Sussex, Brighton, United Kingdom, **4** Centre for Psychedelic Research and Centre for Complexity Science, Department of Brain Sciences, Imperial College London, London, United Kingdom, **5** Centre for Eudaimonia and Human Flourishing, University of Oxford, Oxford, United Kingdom, **6** Department of Clinical Neurosciences and Division of Anaesthesia, University of Cambridge, Cambridge, United Kingdom, **7** Leverhulme Centre for the Future of Intelligence, University of Cambridge, Cambridge, United Kingdom, **8** Montreal Neurological Institute, McGill University, Montreal, Canada, **9** Department of Psychology, University of Cambridge, Cambridge, United Kingdom, **10** Department of Psychology, Queen Mary University of London, London, United Kingdom

‡ These authors are joint senior authors on this work.
* a.proca22@imperial.ac.uk

**Data Availability Statement:** All data and code is available at the public repository: https://github.com/aproca/PID_multitask.

## Abstract

Striking progress has been made in understanding cognition by analyzing how the brain is engaged in different modes of information processing. For instance, so-called *synergistic* information (information encoded by a set of neurons but not by any subset) plays a key role in areas of the human brain linked with complex cognition. However, two questions remain unanswered: (a) how and why a cognitive system can become highly synergistic; and (b) how informational states map onto artificial neural networks in various learning modes. Here we employ an information-decomposition framework to investigate neural networks performing cognitive tasks. Our results show that synergy increases as networks learn multiple diverse tasks, and that in tasks requiring integration of multiple sources, performance critically relies on synergistic neurons. Overall, our results suggest that synergy is used to combine information from multiple modalities—and more generally for flexible and efficient learning. These findings reveal new ways of investigating how and why learning systems employ specific information-processing strategies, and support the principle that the capacity for general-purpose learning critically relies on the system's information dynamics.

## Author summary

What is the informational basis of learning in humans, animals, or, indeed, artificial neural networks (ANN)? Furthermore, how can these systems learn to solve multiple tasks simultaneously? These fundamental questions are, surprisingly, still not fully understood. One advantage of studying ANNs is that we can precisely probe learning-related changes.

**Funding:** This work was supported by the Imperial College London President's PhD Scholarship to AMP (https://www.imperial.ac.uk/study/fees-and-funding/postgraduate-doctoral/grants-scholarships/presidents-phd/); by the Ad Astra Chandaria Foundation to FER; by the Gates Cambridge Trust to AIL (https://www.gatescambridge.org/) and by the Wellcome Trust (grant no. 210920/Z/18/Z to DB, https://wellcome.org/). The funders had no role in study design, data collection and analysis, decision to publish, or preparation of the manuscript.

**Competing interests:** The authors have declared that no competing interests exist.

Here we draw on a recent branch of information theory, partial information decomposition, to examine how *different types* of information support different learning goals, and where, in ANNs. We show that adding noise to an ANN encourages it to keep copies of information at multiple nodes, promoting robustness. In contrast, whenever flexible learning is required, for instance when facing varied stimulus types or diverse tasks, individual neurons work together to represent information more abstractly. This work sheds light on how systems encode information differently according to their learning pressures, which can help us better understand how and why the human brain uses particular forms of information processing.

## Introduction

A central goal in cognitive neuroscience is to understand how the brain processes information to learn and behave intelligently; and a central goal in machine learning research is to recreate these processes on a computer. Historically, this close partnership between cognitive neuroscience and machine learning has been a fruitful symbiosis [1]. Although artificial neural networks are not perfect models of biological neurons [2], they are a valuable tool to investigate how groups of neurons collectively represent and manipulate information [3]. Overall, the aim of this interdisciplinary research effort is not so much to clarify the implementation details of a particular instantiation of successful distributed information-processing (e.g., the human brain), but to extract fundamental principles to allow a better design of a wide range of novel cognitive systems [4].

Information theory provides an ideal conceptual framework for the study of distributed information processing, motivated by the goal of understanding how groups of neurons store, transfer, and modify information [5]. One particularly relevant tool for the analysis of such processes is the recent framework of *Partial Information Decomposition*, or PID [6], that distinguishes the information held by a set of sources about a target variable into qualitatively different components: unique (present in exactly one source), redundant (provided by multiple sources separately), and synergistic information (only available when considering multiple sources jointly). Recent work has revealed a strong relationship between human high-level cognition and synergistic information processing taking place in the so-called 'central executive network,' which involves the lateral prefrontal and parietal cortices, while redundant information has been found to dominate in cortical areas responsible for perception and low-level processing [7]. Synergy and integrated information, of which synergy is a constituent component [8], have also been found to dominate in complex information processing taking place within cellular automata [9,10], and its disruption has been associated with loss of consciousness [11] and ageing [12]. However, despite these promising findings, the precise nature of the underlying mechanisms resulting in the emergence of synergistic information and its utility for computation remains unknown.

In addition to its application to neuroscience, PID is also rapidly gaining traction in the field of machine learning. While the information bottleneck theory is one of the most well-known applications of information theory to neural networks [13–16], the studies used Shannon mutual information which does not capture additional interactions between variables. PID now offers an elegant framework for studying distributed information processing and interactions between parts of a system, which becomes increasingly important as the field moves towards further developing theory of deep learning, better interpreting existing models, and designing new architectures and algorithms that may employ particular information

processing strategies. Furthermore, this approach offers the possibility of understanding of neuronal cooperation and organization in networks that are often difficult to interpret. Recent work developed a differentiable PID measure [17,18], which has been used to specify local learning rules optimizing for interpretable information-processing goals in neural networks [19]. Other studies have used PID for feature selection [20], analyzing training dynamics in convolutional neural networks [21], estimating redundancy in factor graphs [22], and studying representational complexity in ANNs [23], among other work [24,25].

Considering that a crucial feature of human cognition is the ability to learn flexibly and generalize across many different—potentially novel—settings, here we hypothesize that synergistic information may be important for such general-purpose learning. Machine learning provides well-suited avenues for testing this hypothesis by investigating the computations associated with, and the possible utility of, synergy. Although current AI models have yet to reach the level of generality of humans, they have recently shown good multitask performance. For example, "Agent 57" can outperform humans across all 57 Atari games [26], and Gato, a multi-modal, multi-task, generalist policy, uses a single network to answer language questions, caption images, play Atari and 3D exploration games, and control a real robot arm [27]. Although little is known about the information-theoretic properties of such networks, here we propose that a closer investigation will provide a better understanding for the role of synergy in general-purpose learning systems.

To develop these ideas, in this paper we employ simple artificial neural networks with several different architectures in both supervised and reinforcement learning settings as a testbed to investigate general information-processing principles related to learning. The main contributions of this work are (i) to propose functional roles played by information decomposition components in learning scenarios, and (ii) to establish a computational basis for the existing evidence of synergy's importance for complex cognition, with a specific relation to general-purpose learning by supporting multi-modal integration.

## Background: Partial information decomposition

One of the central measures of information theory is Shannon's mutual information $I(X;Y)$, which quantifies the amount of information a random variable $X$ provides about another variable $Y$ by measuring the extent to which knowing $X$ reduces the uncertainty about the outcome of $Y$.

Extending beyond the bivariate case, for a set of random variables (*sources*) $X = (X_1,\ldots,X_n)$ and another random variable (*target*) $Y$, the mutual information $I(X;Y)$ can be separated into distinct terms that describe the partial information contributed by subsets of sources about the target (Fig 1B). As described earlier, these PID terms (or *atoms*) can either be unique ($U$), redundant ($R$), or synergistic ($S$) and can be computed as described in the methods. For the case with two sources, mutual information can be decomposed as

$$I(X_1; Y) = R(X_1, X_2; Y) + U(X_1; Y) \tag{1}$$

$$I(X_2; Y) = R(X_1, X_2; Y) + U(X_2; Y) \tag{2}$$

$$I(X_1, X_2; Y) = R(X_1, X_2; Y) + U(X_1; Y) + U(X_2; Y) + S(X_1, X_2; Y) \tag{3}$$

Although PID proposes the distinction of unique, redundant, and synergistic information, it does not prescribe a unique method for computing these measures and to date there is no single widely-accepted method of doing so. Consequently, many different PID measures have

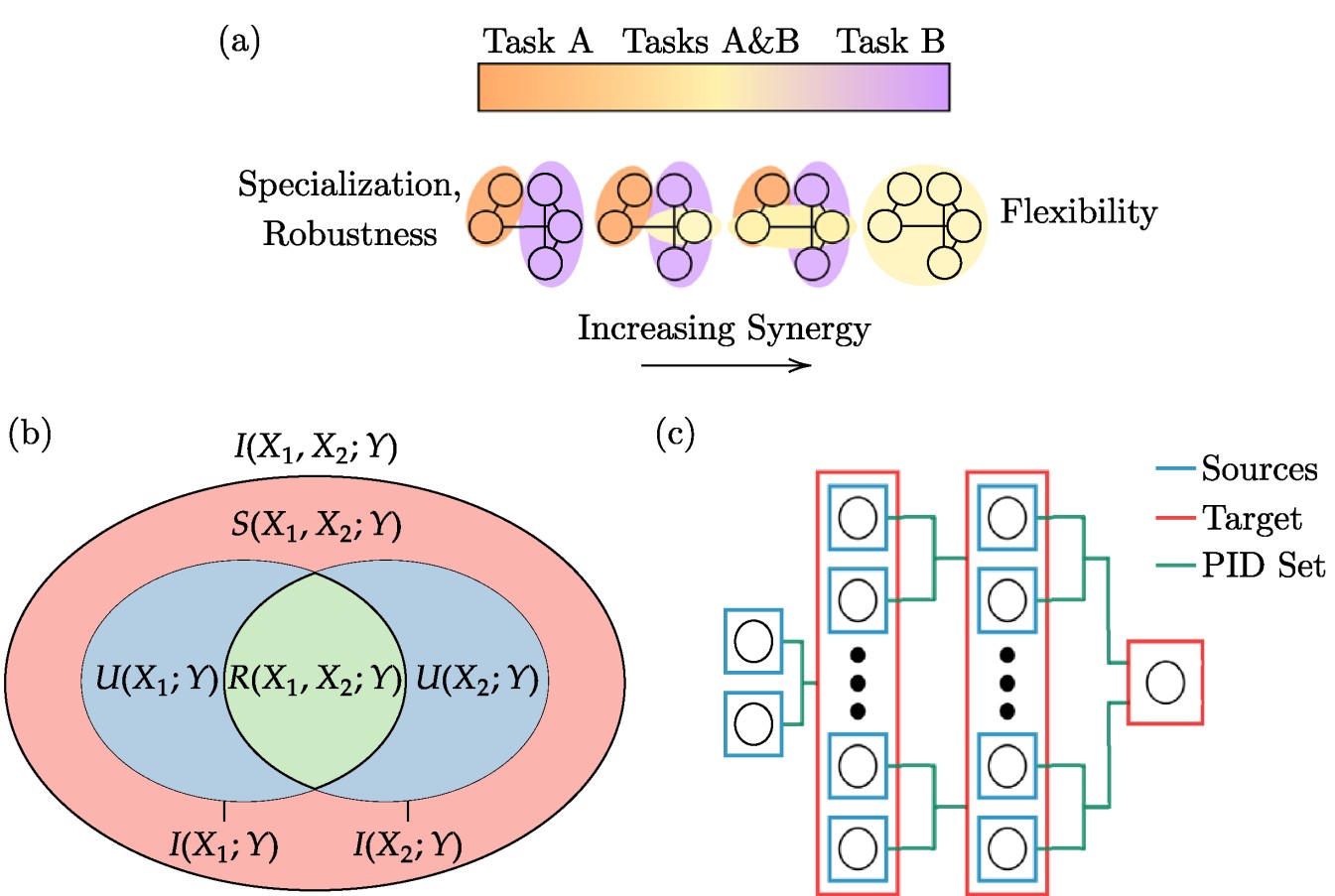

**Fig 1. Information decomposition in neural networks.** (a) A network solving two different tasks could potentially represent information about the two tasks in several ways: it can use distinct populations of neurons for each task (orange, purple) or have some combination of overlapping neuron populations of neurons (yellow) used for both tasks. (b) The decomposition of mutual information between two sources ($X_1$, $X_2$) and a target ($Y$). (c) Example of the set of sources and target considered at different orders in a neural network. The neurons in one layer comprise the set of sources for the target in the subsequent layer. In this setting, the entire layer of neurons is considered collectively as the target. Full-order PID refers to treatment of the entire set of neurons as the set of sources, whereas 2nd-order PID treats pairs of neurons as the set of sources of which all possible combinations are computed and averaged.

been formulated which capture different aspects of multivariate information, some which have been shown to provide a specific operational interpretation (e.g. the payoff in a suitably defined game [28] or the compression through a suitably defined channel [29]). Although past work has shown that in practice different PID measures tend to yield similar results [7,29], it has also been shown that there are situations where different PID can strongly disagree [30]. Taking these considerations into account, we compute all measures using two different redundancy functions, $I_{\min}$ [6] and $I_{\mathrm{MMI}}$ [31], and a range of estimator parameters, to ensure our results are consistent. Although these do not have a meaningful operational interpretation, there are two reasons that make them very suitable for our study: 1) they have previously been used to study neural data which we aim to explain using artificial neural networks; and 2) unlike most other measures, $I_{\min}$ and $I_{\mathrm{MMI}}$ are more pragmatically useful, since they allow us to efficiently estimate synergy in systems with potentially many sources without computing the full PID lattice (mathematically, in both cases the inclusion-exclusion principle can be applied to yield a measure of "union information" in close form). For additional discussion on the field of PID measures, we point the reader to the review of [32].

Consider now a neural network learning two distinct tasks. The network can represent these tasks in different ways. It can assign a particular set of neurons to one task, and a separate set to the other (orange and purple neurons in Fig 1A). Alternatively, it could use the same overlapping set of neurons to encode information about both tasks, distinguishing tasks by the collective behavior and interactions of such neurons (yellow neurons in Fig 1A). The first method specializes its neurons by designating them for particular tasks, whereas the second method reuses its neurons for multiple tasks. In particular, the first method uses unique (and potentially redundant if the same information is provided by several neurons) information to solve the two tasks, and the second method uses synergistic information. Intuitively, the first approach conveys specialization and robustness, while the second approach provides potentially greater flexibility and reusability, as also suggested in prior work [33].

Mutual information can be decomposed differently at different scales and is dependent on the selection of sources and targets. For example, each task-specific group of neurons can also vary in its decomposition, such that the group solving Task B could be more synergistic than the group used for Task A—even though the neuronal populations across tasks do not overlap. These types of scenarios can be studied by considering different sets of sources in a network over which to compute PID (Fig 1C): one could use all neurons in a layer as a single set of sources and the joint state of the next layer as the target (full-order), or select all the combinations of pairs of neurons in a layer as sets of sources and average the resulting information atoms ($2^{\text{nd}}$-order), or anything in between. Studying different scales can contextualize information-processing behavior in terms of the interactions occurring between different sets and subsets of system parts. With the conceptual framework of PID and the numerical estimators presented in the methods, one can properly investigate the information decomposition of neural networks, to which we now turn our attention.

## Results

We now present a series of experiments exploring the properties of synergy and redundancy in small neural networks with two hidden layers of ten neurons each. First, we look at information flow in networks solving logic gates, where the types of information needed to solve the task are known. We then study a more complex setting in which logic gates are embedded and extended in a 3D simulated environment and solved using reinforcement learning, either individually or in a set of tasks. Finally, we explore networks learning multiple tasks further using the NeuroGym suite of tasks inspired by cognitive neuroscience experiments [34]. Overall, our experiments converge on synergy being associated with multi-modal integration and the learning of multiple tasks, and redundancy with robustness.

### Functional roles of information atoms in simple learning problems

We first study redundancy and synergy in small feedforward networks learning a copy (COPY) or exclusive-or (XOR) logic gate involving two inputs (Fig 2A), which are well-defined tasks with known informational requirements and few confounding factors. In particular, the processing done by the COPY gate involves no synergy, as it requires no integration of information as it is solved by simply copying the first dimension of the input. Conversely, an XOR gate is solved by integrating information from both dimensions of its input as it reflects the parity of the set (i.e., if the inputs are similar or different). In fact, the XOR gate is known to be maximally synergistic [35], as there is no reduction of uncertainty about the output unless all input sources are considered jointly. We perform analyses at both the $2^{\text{nd}}$-order and full-order scales to compare how information profiles vary depending on the number of sources considered (Figs A and B in S1 Text); a discussion on our choice of order can be found

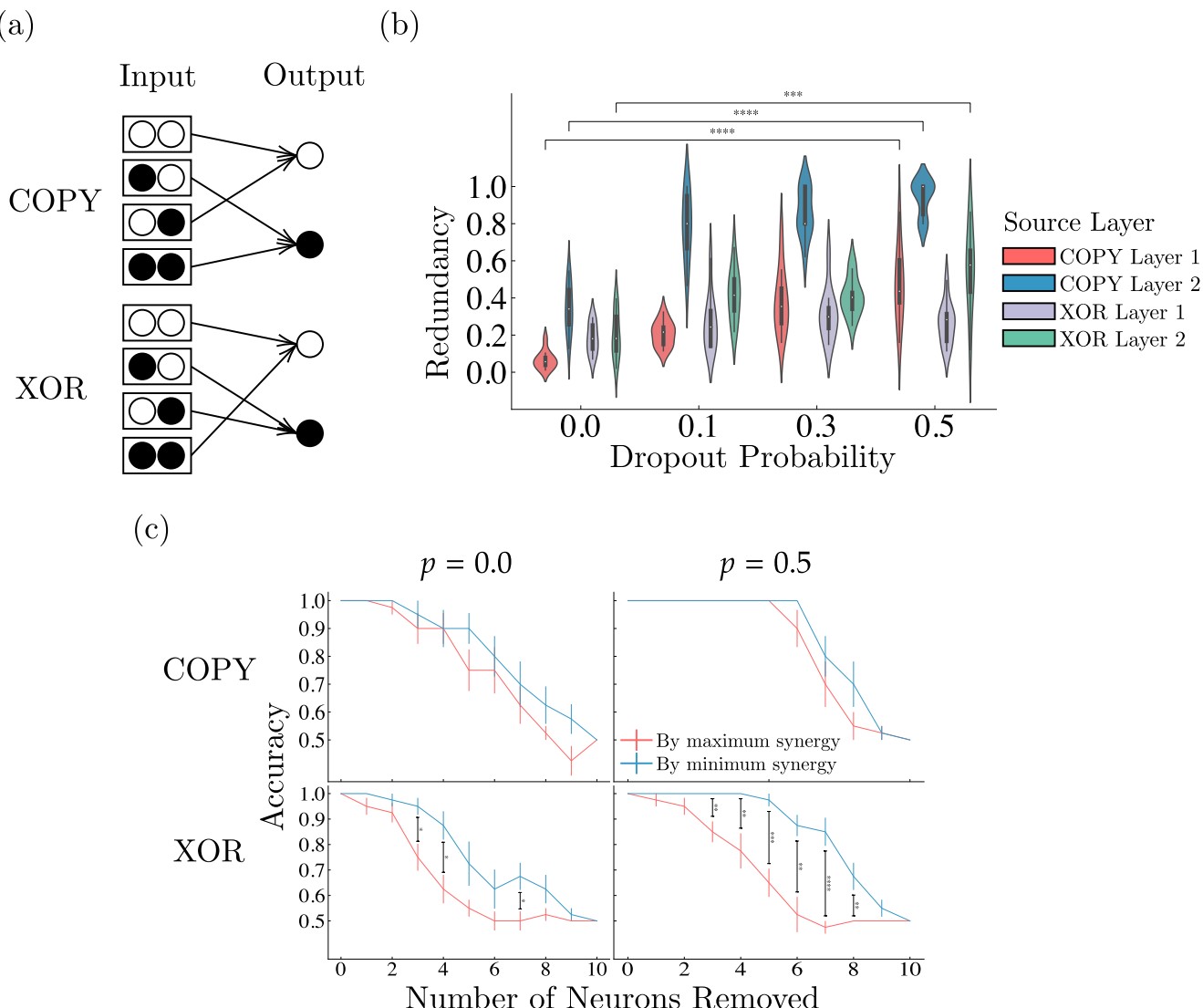

**Fig 2. Effects of lesions and dropout on network information profiles.** (a) White nodes represent 0 and black nodes represent 1, specifying the data used for each logic gate. (b) Dropout increases redundancy across the hidden layers of the network for both COPY and XOR tasks (***$P<0.001$, ****$P<0.0001$, independent samples $t$ test; $n = 20$). By forcing the network to decrease its reliance on individual (sets of) neurons by randomly turning them off during training, dropout encourages redundancy to overrepresent important information. Values represent probability density functions. (c) Lesioning by permanently removing neurons during evaluation shows the causal role of synergy-rich neurons for task-performance, especially for tasks requiring integration of information (XOR) (*$P<0.05$, **$P<0.01$, ***$P<0.001$, ****$P<0.0001$, paired samples $t$ test with Benjamini-Hochberg False Discovery rate correction; $n = 20$). By increasing redundancy, dropout (denoted as $p = 0.0$ and $p = 0.5$) results in decreased reliance on individual neurons, allowing the network to be more robust to their removal. For the XOR gate, which requires the integration of information to solve, even after applying dropout performance quickly degrades when synergistic neurons are affected; this loss in performance is substantially attenuated if non-synergistic neurons are removed. Values represent means ± SEM.

in the methods. We refer to 2^nd^-order measures in text and figures unless otherwise specified. We also perform the same experiments using networks with twice the number of neurons to ensure our results are robust across different hidden layer sizes (Figs C-E in S1 Text). Finally, we perform an analysis comparing the decomposition of different layers of the network (Figs U and V in S1 Text; see methods).

**Dropout removes irrelevant input information and increases hidden layer redundancy.** To study how learning pressures can influence the information profile of a network,

for each logic gate we apply four levels of dropout ($p = 0.0, 0.1, 0.3, 0.5$), a popular regularization method in deep learning that randomly omits different neurons (with some given probability) in each forward pass during training. This differs from "lesioning" in that it is applied during training, and a neuron is not permanently rendered inactive. This forces the network to be robust and able to adapt to random perturbations applied to its neurons, rather than strongly dependent on particular sets of neurons. Furthermore, by applying dropout, we disrupt synergistic and unique information because they rely on individual neurons that may be randomly "turned off," while redundant information persists in other neurons that remain "on"—making it an interesting paradigm for investigating the resulting learned information profiles.

By applying increasing amounts of dropout, we find that redundancy and synergy from the input significantly decrease for networks solving COPY gates, which rely only on the first input dimension and learn to ignore the second dimension, but are preserved for XOR gates, which require both dimensions to successfully complete the task (Fig F in S1 Text). Thus, dropout encourages the removal of task-irrelevant information from the input. This may be due to the increased risk of information interference dropout yields—if an important neuron which modulates the input from an unimportant neuron is removed, the network could be influenced in a disadvantageous way.

Dropout also causes networks to over-represent important information. As shown in Fig 2B, applying dropout significantly increases redundancy in the hidden layers for both logic gate tasks. As the risk of a neuron's removal increases, the network must compensate by ensuring that a robust, redundant representation of important information remains, decreasing the reliance on individual (sets of) neurons (i.e., unique information). Thus, with limited information resources and the heightened risk of information loss, task-irrelevant information is more likely to be removed in favor of task-relevant information, which is instead over-represented. Our finding provides an explicit measure of redundancy, complementing recent work [36] that has also suggested dropout to increase redundancy based on an increase in clustering driven by the similarity of neurons.

**Performance relies on synergistic neurons.**   Using the trained logic gate networks, we perform lesioning experiments to evaluate whether a neuron's synergy is predictive of its importance to the network. In lesioning experiments, each neuron's pairwise synergy (average synergy with every other neuron in the same layer) is computed. We then permanently, iteratively remove the most (or least) synergistic neurons from each layer by setting their outgoing activations to 0 and evaluate subsequent performance. This differs from dropout in that lesioning is performed after training during evaluation, rather than during training, such that the network is unable to modify its parameters. By using networks with and without dropout applied, we can observe how dropout, and its resulting increased redundancy, influences the reliance on synergistic and non-synergistic neurons for performance.

Lesion experiments reveal that synergistic neurons are more critical for performance than non-synergistic neurons (Fig 2C), especially for tasks requiring the integration of information (e.g., XOR). Synergistic neurons have less robust representations—the removal of one synergistic neuron can change the information carried by all of the sources acting synergistically with it, while the same is not true for redundant or unique information. Thus, synergy-rich neurons are more sensitive and their removal decreases performance more than synergy-poor neurons.

These results further show the effects of dropout on increasing robustness of the network via increased redundancy: with higher dropout, more minimally-synergistic neurons can be removed without disrupting performance as their information is overrepresented through an increase in redundancy. However, the XOR logic gate networks still remain highly sensitive to a disruption in synergistic neurons even after dropout is applied, exemplifying the importance

of synergy for tasks requiring integration, as well as the vulnerability of such representations. Conversely, because the COPY logic gate networks do not need to integrate information in order to solve the task, dropout instead reduces the number of synergistic neurons and the reliance on them such that their removal does not disrupt performance until over half of their neurons have been removed.

## Compositional tasks in 3D RL environment

As a second step in our investigation, we extend the idea of solving logic gates to the context of reinforcement learning agents in Animal-AI [37], a 3D environment with simulated physics used for assessing agents on cognitive common-sense physical reasoning tasks [38] (experimental details can be found in the Methods). These experiments are motivated by the interest in studying synergy in the context of task-transfer and how agents may allocate their parameter space when learning a novel task after specializing on an initial task—and, more broadly, on how the structure of new tasks could influence information decomposition. In effect, contrasting with the previous tasks in which training happens directly on the desired input and output, agents in these reinforcement learning scenarios learn through trial-and-error interactions with their environment while trying to maximize a reward function. This significantly increases the difficulty of the task as reward is delayed, the environment contains a much larger state space, and the observation space includes additional (task-irrelevant) inputs about the environment. With these added challenges, the agent must use information about the arena walls to determine which platform direction (forward or backwards) to move to and act accordingly in order to retrieve the positive reward and solve the task.

**Information profiles reflect specific task demands.**   We train models to perform either an individual task or a set of tasks in sequence, usually denoted in AI research as a *curriculum*. Each task consists of several environment configurations (each corresponding to a configuration of logic gate inputs) that are interleaved across episodes. In the curriculum experiments, models are trained on each task until reaching a reward threshold or a maximum number of training steps, after which they are trained on a new task.

Each considered task follows a similar design in which the agent aims to solve a problem based on the object-type of the barriers surrounding it (Fig 3). Specifically, the agent receives as input three raycasts (indicating the type of the objects to its front, left, and right), and as output the agent can move either forward or backward into a pit to obtain a reward. The object-type of the barriers encodes three input bits (wall being 0, cardbox being 1), and the position of the reward encodes the correct output (forward being 0, backward being 1). To successfully solve the task, the agent must decide based on the barrier types which direction to move, in order to retrieve the positive reward. The direction of the reward corresponding to the configuration of the barrier types is determined by the logic gate task being performed, which includes the same gates as in the previous experiments (2-Bit COPY and 2-Bit XOR; Fig 3B, plus a 3-Bit extension of the XOR gate (where the correct output is the parity of all inputs; Fig 3C), and a "Distance XOR" task where the length of the platform is increased, introducing a longer delay between action and reward (Fig 3D).

Our results show that networks increase their synergy as they learn new tasks, and that the integration of an additional source of information specifically drives this behavior (see Fig 4A). Synergy significantly increases from the 2-Bit XOR task to the 3-Bit XOR task, despite agents not being able to learn the second task to perfect accuracy, whereas synergy remains constant across all Distance XOR tasks (Fig 4B), even when learned accurately. Although the 3-Bit XOR alone does yield more synergy than the 2-Bit XOR task alone, even when the 3-Bit

## 2-Bit XOR

## 3-Bit XOR

## Distance XOR

**Fig 3. Animal-AI tasks.** (a) An image visualization of the Animal-AI environment [37] from the position of the agent for the '01'-input 2-Bit XOR configuration, where the '0' gate output corresponds to a reward situated in the pit behind the agent. The agent receives raycast observations (rather than pixel inputs) of the environment from its position, which occlude the content of the pits. (b) An aerial view of the '10'-input 2-Bit XOR configuration, where the left cardbox barrier represents a '1,' the right wall barrier represents a '0,' and the backward-relative-to-agent position of the green reward represents a '0' output of the gate. The orange lines represent the orientation of the raycasts projected from the agent. (c) The configuration of the 3-Bit XOR task for '001'-input, where the additional barrier in front of the agent represents a third logic gate input. (d) Example configurations (each part of a separate task corresponding to platform-length) for the Distance XOR set.

XOR task is not accurately learned in the curriculum, the learning process of the new task drives an increase in synergy.

Conversely, the Distance XOR curriculum does not drive an increase in synergy, although agents successfully learn all tasks. The main factor distinguishing the 2-Bit to 3-Bit XOR curriculum from the Distance XOR curriculum is that the former requires integration of an additional source (3 bits instead of 2). Instead, the latter requires learning new tasks, but does not require the incorporation of any new sources of information or any modification in its processing—only the association of more states with a particular learned action using the same mapping from existing sources. Overall, these differences in synergy between curricula highlight a difference in the complexity of the set of tasks being learned, as sets of tasks that are more simple (e.g., using few sources of information in a specialized manner) do not drive an increase in synergy, while more complex tasks dependent on the integration of several sources do. This suggests that synergy is specifically related to the learning of multiple complex tasks in which multiple information sources have to be integrated to yield novel behavior. Finally, we compare the level of synergy in different layers of networks (Figs W and X in S1 Text; see Methods).

### Effect of learning multiple diverse cognitive tasks on synergy

Our final set of experiments seeks to analyze networks learning multiple cognitively-inspired tasks, and investigate how learning a set of tasks requiring the capacity to integrate different modalities compares to a set of tasks relying on a single modality. When trained on a non-stationary sequence of tasks (for example, a sequential curriculum), neural networks often suffer

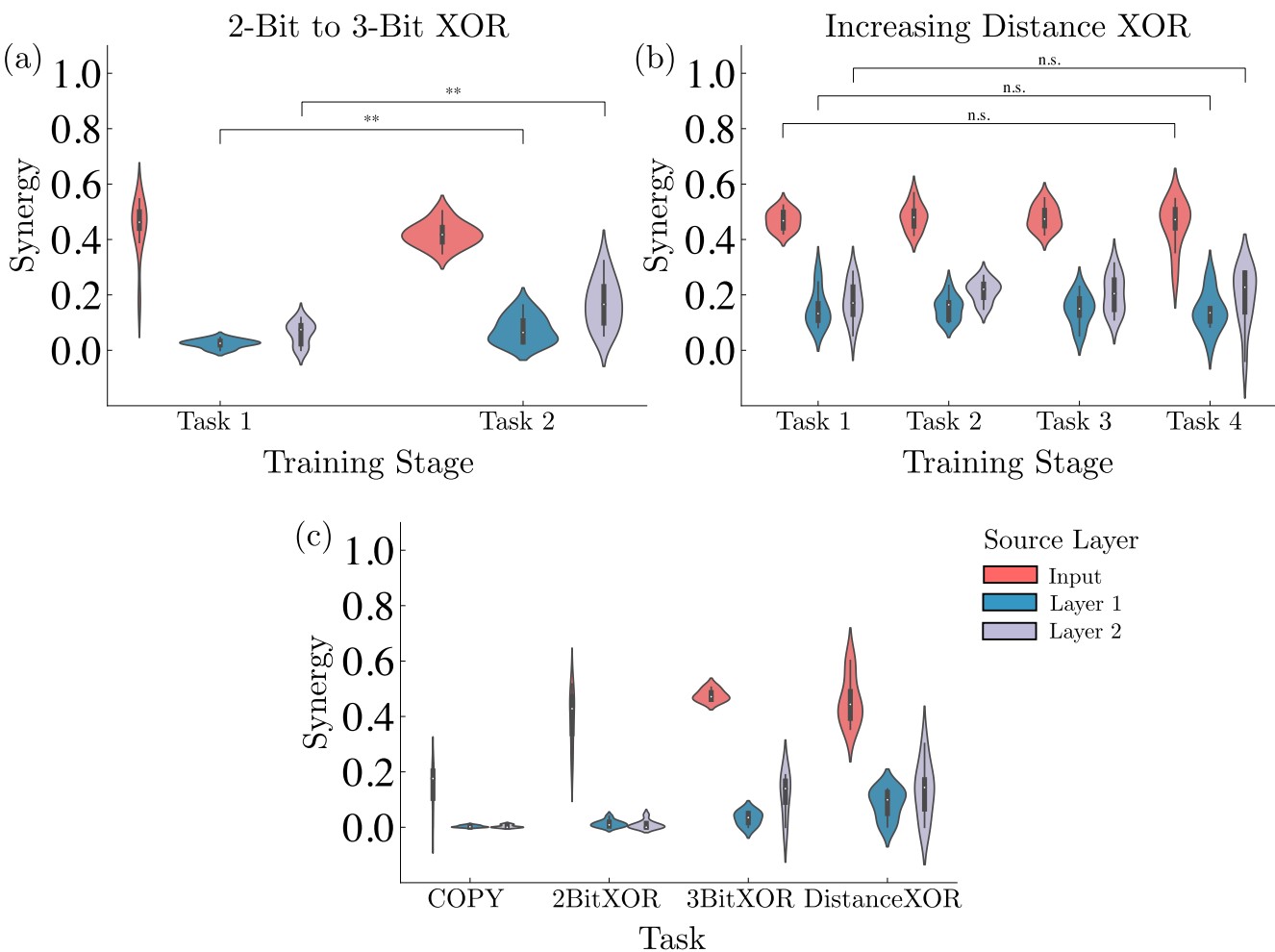

**Fig 4. Relation of compositional tasks and synergy in Animal-AI.** (a) Synergy significantly increases from the end of training on the 2-Bit XOR task to the end of training on the 3-Bit XOR task, driven by the addition of a third source of information to integrate (**$P<0.01$, paired samples $t$ test; $n = 20$). Values represent probability density functions. (b) Synergy does not significantly increase in the Distance XOR curriculum (n.s., not significant, paired samples $t$ test; $n = 20$), likely due to the fact that each subsequent task requires the same integration of sources and only varies in the number of times an action must be performed to reach the reward. This could be done easily by ignoring global position and object distance or by extending the number of environment states associated with an action, without additional integration of information. Values represent probability density functions. (c) The information decomposition of a network is influenced by the task it's trained on and is qualitatively different across tasks. Distance XOR refers to Distance 10 XOR. Values represent probability density functions.

from interference as they update parameters to learn a new task that potentially disrupt setups that were important for solving previous tasks, resulting in a phenomenon known as 'catastrophic forgetting' [39]. Analyzing networks that can remember (rather than forget) their entire training curriculum allows us to study how information profiles are influenced by learning and solving multiple (versus single) tasks. With this motivation, we use two different training protocols for each pair of tasks: sequential training, which does not prevent forgetting the first task trained on, and interleaved training, which forces the network to retain the capacity to solve both tasks.

**Studying RNNs learning decision making tasks.** We explore information decomposition in the hidden layer of recurrent neural networks (RNN)—a simple form of memory—on tasks requiring integration over time. This model is used to solve tasks taken from the NeuroGym

environment [34], a toolbox of cognitive neuroscience tasks for training ANNs. Previous work has used this framework to compare ANNs to animal studies, for example in studying population gating of task-relevant features in the primary auditory cortex of rats [40], and replicating suboptimal behavior produced by structural priors in rats [41]. We use a subset of their collection of decision-making tasks, referred to as the 'DM family' (DM1, DM2, CtxDM1, CtxDM2) in [42]. This set of tasks is based on various decision-making tasks used in neuroscience and psychophysics [43–45], involving the presentation of two simultaneous stimuli (e.g., numeric value of the input at a particular dimension) within a specific modality (set of input dimensions; e.g., DM1 presents stimuli in modality 1 and DM2 in modality 2), after which models must indicate which stimulus is stronger (higher in value). In contextual decision-making tasks (CtxDM1, CtxDM2), a second set of stimuli is also presented in the other modality (the remaining set of input dimensions) and must be ignored. In non-contextual tasks, stimuli are not presented in the other modality. We refer the reader to [34,42] for a detailed description of all aforementioned tasks. For each task, a set of consecutive trials is given as input to the RNN. Each trial consists of an initial fixation phase, followed by stimuli, and ending with a decision-making period. To successfully solve a task, the model must integrate stimulus information over time to make its final decision.

To better understand the relationship between synergy and the curriculum of tasks being learned, we compare networks trained on pairs of tasks we define to be congruent or incongruent. In this context, congruence refers to the similarity between both tasks and whether learning one task may aid in learning of the other task (such that transfer of performance is possible). In our experimental design, congruent tasks are defined as the pairing of decision-making tasks using the same modality (DM1&CtxDM1; DM2&CtxDM2) and incongruent tasks are those using different modalities (DM1&DM2; DM1&CtxDM2; DM2&CtxDM1; CtxDM1&CtxDM2). Learning congruent tasks requires attending to and integrating information from a single input modality and ignoring stimuli in the other modality, while learning incongruent tasks requires switching between attending to one and ignoring the other of two modalities, depending on the task. We further compare networks learning a curriculum of tasks sequentially and networks learning both tasks simultaneously through interleaving (Fig 5A). We do not use any continual learning methods to prevent catastrophic forgetting in either case. Therefore, sequential learning presents a condition where forgetting a previously experienced task could potentially occur while interleaved learning forces the network to solve both tasks.

**Solving diverse sets of tasks increases synergy.** Our results show that networks trained on congruent tasks yield similar levels of synergy and accuracy in both sequential and interleaved training regimes (Fig 5B). The similarity of congruent tasks allows for easy transfer, as their learned parameters can be reused across tasks without strong interference, achieving comparable performance in both sequential and interleaved protocols. Because both tasks are similar and only require attending to and integrating from a single modality, networks trained sequentially can reuse their representations for both tasks and achieve adequate performance, without accommodating additional information. We suggest that because two congruent tasks require information to be integrated from only a single modality, levels of synergy are lower for all training settings than for two incongruent tasks.

In contrast, networks trained on incongruent tasks fall in different clusters depending on the training regime (Fig 5C). In effect, interleaved training yields clusters with higher synergy and accuracy, whereas sequential training yields clusters with lower synergy and accuracy. This occurs because sequential training of incongruent tasks leads to some forgetting of the first task due to interference, leading to lower accuracy, while interleaved training forces the network to solve both tasks resulting in higher accuracy. Furthermore, networks trained with

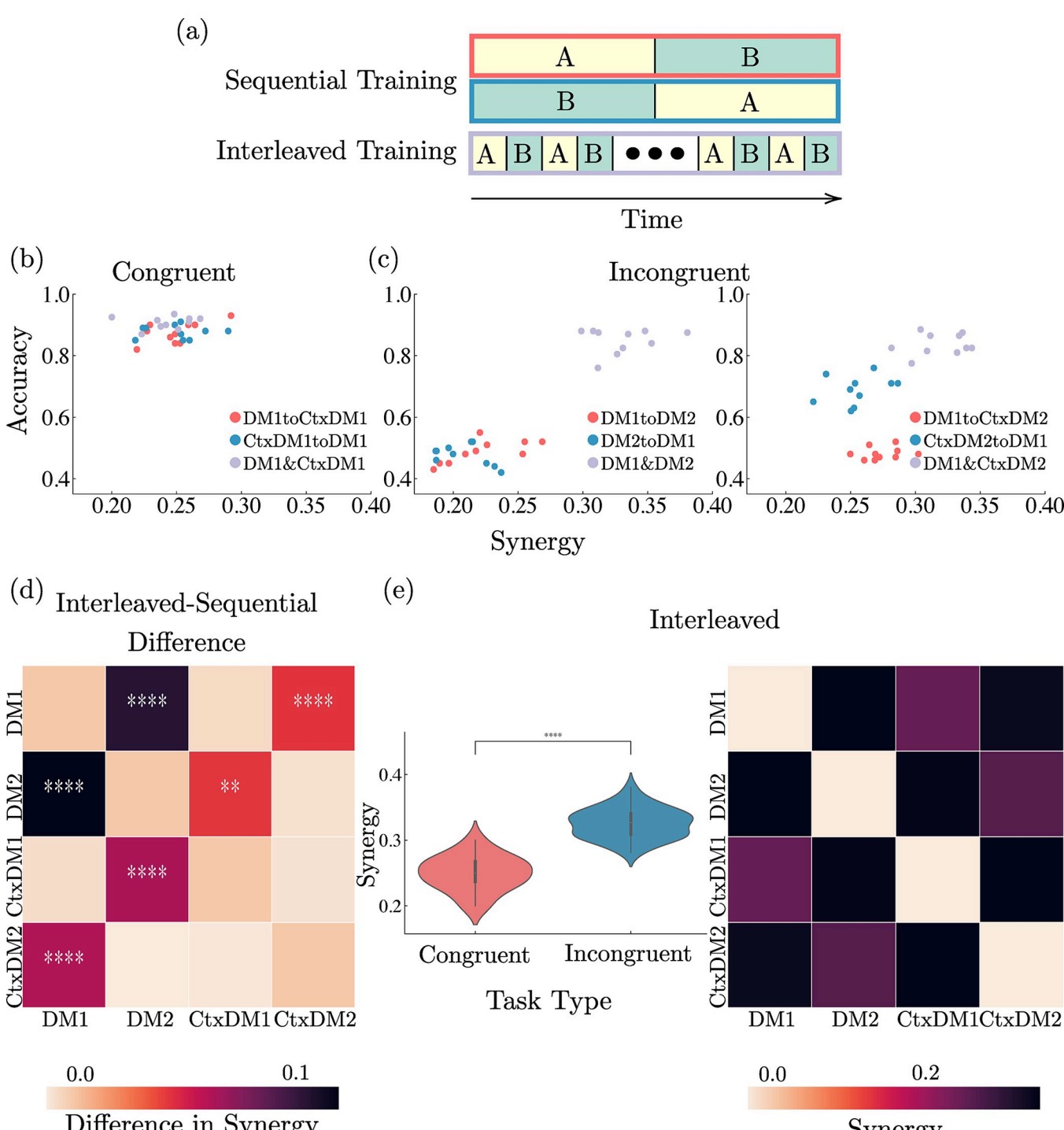

**Fig 5. Synergy increases with the solving of incongruent tasks.** (a) In the sequential protocol, networks are trained on each task in a single block sequentially (red and blue in panels b-c) and forgetting can occur. In the interleaved protocol, networks are trained on both tasks simultaneously (orange in panels b-c) and thus are both solved. (b) Congruent tasks yield similar accuracy and synergy for both training protocols. Values represent individual data points. (c) Incongruent tasks yield distinct accuracy-synergy clusters based on training protocol. Sequential training yields lower synergy corresponding to worse performance, while interleaved training performs better with higher synergy. Values represent individual data points. (d) Training with interleaving yields significantly higher synergy than training sequentially for incongruent tasks, but not for congruent tasks (**$P<0.01$, ****$P<0.0001$, independent samples $t$ test with Benjamini-Hochberg False Discovery rate correction; $n = 20$). Values represent the difference of means of networks trained with interleaving versus sequentially. (e) Networks accommodating two incongruent tasks (via interleaving) yield significantly higher synergy than those accommodating two congruent tasks (****$P<0.0001$, independent samples $t$ test; $n = 60$). Left: values represent probability density functions. Right: values represent means.

interleaving have an increased capacity to integrate information from both modalities in order to adequately perform both incongruent tasks, resulting in a higher level of synergy compared to networks trained sequentially.

Overall, in networks trained with interleaving, the amount of synergy is significantly higher for incongruent tasks compared to congruent tasks, and is consistently higher for individual pairs of incongruent tasks than pairs of congruent tasks (see Fig 5E). This suggests that the capacity to integrate and use information from several modalities (as in incongruent tasks) results in a higher proportion of synergistic information compared to that yielded by a single modality (as in congruent tasks). In other words, synergy is related to a system's ability to combine different sources of information flexibly for distinct tasks. We speculate that this insight may help to explain recent empirical findings showing a higher synergy associated with the brain's associative cortices, which integrate information from multiple sensory systems, in contrast to unimodal brain areas such as sensory or motor cortices [7].

Furthermore, by contrasting networks with interleaved vs. sequential training we can attribute this synergy increase specifically to the capability of simultaneously solving multiple incongruent tasks (Fig 5D). Sequential training of incongruent tasks results in the forgetting of the first task and a corresponding drop in performance as well as a drop in synergy (compared to interleaved training), which is related to the fact that the network is only using information from one modality at a time. This doesn't hold for congruent tasks, however, for which both interleaved and sequential training result in similar levels of both performance and synergy. This results in synergy being specifically linked with scenarios where the network needs flexibility to adapt between tasks requiring integration of different modalities—and, in these cases, synergy is highly correlated with performance.

## Discussion

This paper presents a series of experiments using neural networks in a variety of tasks and learning settings, and examines their internal representations using the Partial Information Decomposition (PID) framework [6]. Based on our results, we draw several interpretations of the functional roles played by different forms of information, and suggest their relation to learning in artificial and biological neural networks.

### Functional roles of information atoms

We start by delineating the functional roles of redundancy, unique information, and synergy in learning contexts.

**Redundancy.** Although redundant information makes less efficient use of neuronal capacity, it grants robustness to the network. In effect, over-representing important information (i.e., encoding it in multiple units) is an effective way to ensure that it will be propagated through the network, even in the presence of possible interferences. Applying perturbations to the network during training—such as dropout—incentivizes such reliability, which naturally leads to an increase in redundancy. Furthermore, after training with dropout, these networks resist performance drops caused by lesions to a much higher degree than networks trained without dropout. In the human brain, redundancy dominates lower-level cortical regions, particularly in sensory and motor areas [7]—which could be due to a similar need to over-represent important sensory information in order to extract critical features of incoming data and resist perturbations (e.g., noise).

These findings have interesting parallels with previous analyses of neural networks based on the information bottleneck principle [13], which uses mutual information to bound optimal networks trading off information loss due to compression and information preserved

about a desired output [14–16]. In agreement with that line of work, the results of our logic gate experiments show that dropout acts as an information bottleneck through its increase of redundant information and pruning of task-irrelevant information—the mutual information between the network and the input is decreased (via removal of task-irrelevant features), while the mutual information about the desired output is increased (via higher redundancy). However, our results also reveal phenomena beyond the usual formulation of the bottleneck: our work shows that dropout not only changes the total mutual information, but it also changes the structure of that information by altering its composition. This finding emphasizes the fact that it is not only the content of a task, but also how it is learned, that affects the information processing strategy adopted by a cognitive system.

Our interpretation of dropout provides a complementary perspective to the classic view of dropout as preventing complex co-adaptations [46,47]. Rather than learning complex, distributed 'co-adaptations,' which may link correlations in the input between task-relevant and task-irrelevant information, dropout effectively sparsifies the network by increasing the risk of losing information, causing the network to learn to overrepresent task-relevant information (redundancy) while decreasing information coded across many neurons, which has a higher risk of being disrupted (reduced synergy), and to discard task-irrelevant information which may interfere.

**Unique information.** Unique representations provide specialized encoding of information, whereby encoding can take place in a single neuron rather than requiring a set of neurons to operate as a distributed representation (as is the case for synergy). Such specialized representations are particularly efficient when a network consistently performs the same task or several tasks with the same substructure, or when it is not required to integrate multiple sources of information.

We speculate that the utility of unique information could be harnessed by functionally specialized circuits yielded by evolution and early development, especially in sensory cortices. The one-to-one mapping of simple receptive fields in the primary visual cortex is a clear example of such specialization in the brain. High-level cognition, in contrast, requires the integration of information from several cortical regions being in principle more diverse and less stable than processes associated with low-level feature extraction. Nevertheless, regardless of what cognitive process is occurring in higher cortical regions based on visual input, the visual cortex remains a specialized and stable region for the subtask of visual feature extraction, rather than other sensory functions. For these types of tasks, the flexibility that may be provided by synergistic representations and the associated learning processes are not necessarily beneficial. The empirical evaluation of these conjectures is an interesting avenue for future investigations.

**Synergy.** Perhaps one of the greatest functional advantages of synergy is that it can encode more information than other information atoms for a given population size [29]. In effect, in contrast with redundancy and unique information, synergy relies on combinations of neurons, which makes its informational capacity grow exponentially with system size. Synergistic information, however, is also more vulnerable to noise [10], because a distortion in a single source could disrupt information synergistically held together with other sources. This vulnerability to noise may partially explain why networks exhibit higher levels of synergy at comparatively lower orders rather than higher orders [25] as an attempt to minimize the effect of losing synergistic information with several neurons.

Our experiments find that the removal of maximally synergistic neurons yields a larger drop in performance compared to minimally synergistic neurons. Building on the above discussion, this finding can be explained along three—not mutually exclusive—lines of reasoning: (1) synergistic neurons may be encoding more information collectively than other neurons encode individually with other forms of information, (2) this information is necessary for

integrating several sources of information, and (3) this total information is more vulnerable to perturbations because it can be altered by a disruption in a single source. Future work may seek to disentangle the relevance of each of these potential causes in driving this effect.

## Synergy facilitates flexible general-purpose learning

In addition to the functional roles highlighted above, an important overarching result is the association between synergy and the learning of multiple tasks. Results across all our experiments demonstrate that synergy supports integration in networks solving multiple tasks. Specifically, our logic gate experiments show that performance relies on synergy-rich neurons for tasks requiring the integration of multiple sources of information, and that these neurons are more sensitive to perturbations. Results in the Animal-AI experiments show that the incorporation of additional sources of information when learning multiple tasks drives an increase in synergy, as opposed to learning multiple tasks that do not substantially change the processing of information sources. Thus, the complexity (loosely understood here as relying on integrating various information sources in flexible and diverse ways) of a task set relates to the degree of synergistic information processing it elicits. Finally, results in NeuroGym decision-making tasks show that synergy increases with a network's capacity to simultaneously integrate information from several modalities for different tasks. Altogether, these findings support a link between synergistic information processing and the ability to perform multiple complex tasks, which require the flexibility to integrate and process various sources of information in different ways. In addition, we hypothesize several other functional advantages synergy could provide in learning systems performing several tasks.

The first hypothesis is that, in addition to providing additional capacity for modality integration, synergy could be a response to the learning pressure of having to encode more task-relevant information overall in the neuronal information space—which, in the case of our experiments, is severely constrained by size. Thus, in order to successfully solve two or more tasks requiring the integration of diverse sources, representations may need to be encoded in increasingly efficient manners—a feature which is provided more readily by relying on synergy than unique or redundant information. Performing increasingly complex tasks could have a similar effect as the number of task-parameters and variables increases, and a higher number of information sources and proportion of information must be integrated and encoded in a network. In this way, a complex task could just be seen as a collection of many smaller and simpler tasks.

A second hypothesized utility of synergy is its ability to represent information in a structurally different way. Whereas unique information can provide information along a single dimension for each feature encoded by a source, synergy could provide higher-dimensional representations across several neurons signifying distance between different features and neuronal encodings. This could aid in representing structure and similarity across and within tasks, providing increased flexibility for generalization. Alternatively, synergistic information could be used for integrating other (possibly specialized) representations, potentially occupying some low-dimensional subspace.

## Implications for cognitive and computational neuroscience

Our results and hypothesized functions for synergy in the context of general-purpose learning are complementary to those observed in the brain. Within cognitive science, the division of complex tasks into simpler sub-tasks has been proposed for many decades as an important mechanism for solving almost any cognitive goal [48]. In addition, this parsing of tasks into sub-tasks has been associated with the prefrontal parietal network in brain-scanning

experiments [49–52]. This brain network has been shown to preferentially support synergistic forms of information processing. For instance, [7] found that synergistic information is higher in regions of the brain that are especially responsible for complex human cognition (particularly association cortices in frontal and parietal regions, including the default mode and executive control networks), and that redundant information is higher in areas responsible for perception and low-level cognition (particularly primary motor and sensory cortices). The increased synergy in higher cortical regions may be explained by their need and capacity to integrate information from other brain regions, encode and learn a vast information space throughout life, and reuse or relate this information flexibly in order to generalize to new settings [53–55].

In addition, just as synergy could contribute to creating higher level representations and structure in the context of artificial neural networks, synergistic brain networks tightly overlap with regions that integrate information [11]. Whereas low-level brain regions may benefit from having redundant and specialized representations that are more robust and, correspondingly, comparatively less adaptable after development, higher cortical areas continue to have the responsibility of learning throughout life, reflected in their information decomposition.

In addition to studies featuring information-decomposition analyses of aggregate neural signals, such as fMRI, which initially motivated the current work, other work has used PID to analyze computations at the population and single neuron levels, which offer a more natural analogue to ANNs. One study [56] found that in *in vitro* spiking cortical neurons of mouse somatosensory cortex, greater recurrent information flow (in terms of number of recurrent edges and strength of connectivity) was associated with higher synergy, whereas greater feedback relative to feedforward information was found to result in less synergy. In alignment with the authors' discussion, our study suggests that this may be explained by the importance of recurrent connections for lateral integration of multiple features, which elicits increased synergy, whereas feedback may reduce the variance of feedforward information and its corresponding strength, perhaps performing less integration of distributed sources of information and instead specializing along certain pathways. While other studies have recognized the utility of synergistic processing, many have also highlighted the important role of redundancy in neural computation. For example, noise correlations in the association cortex of mice were shown to predict better task performance despite decreasing sensory information [57]. One explanation offered was that such correlations are the result of redundancy in neural representations used to enhance signal propagation. Although these results seemingly conflict with other evidence of association cortices being highly synergistic, the particular study used Pearson correlation and PCA, both of which cannot reveal the separate contributions of unique, redundant, and synergistic effects, and thus it is difficult to evaluate exactly how these findings translate to the framework of PID specifically. Nevertheless, other work has indeed revealed that increased redundancy is associated with correct behavioral responses related to perceptual discrimination in the mouse auditory cortex [58]. This aligns with our finding that redundancy offers robustness in the presence of noise, and our prediction that such robustness is especially realized in low-level sensory cortices.

Our study of the relationship between ANNs and biological neural networks has interesting parallels to the study of distributed versus localist processing. In particular, parallel distributed processing (PDP) models from psychology [59–61] claim that the brain's encoding of information is distributed, rather than represented locally in specialized representations, and non-symbolic. Indeed, the predecessors of modern deep learning models, PDP models have been successful in helping to explain computation underlying cognition. However, theories about localist processing have found support in ANNs which, although designed to be parallel and distributed, also appear to develop semantically meaningful categories within specialized units.

Our work further unites these two perspectives, as we observe networks strike a balance between specialization and redundancy versus distributed synergistic representations. We suggest that both localist and distributed forms of information processing may be utilized in different settings, depending on the type of computation being performed, noise, and perhaps other factors such as the environmental statistics related to the information being represented.

Related to the current paper, deep learning has been previously used to study representations elicited by networks performing multiple tasks. Prior work has shown that distinct functional units emerge in networks trained on multiple tasks, becoming specialized to specific sub-task features [42], that networks learning multiple tasks naturally produce abstract representations [62], and that networks performing context-dependent tasks use task representations lying on a low-dimensional and orthogonal manifold [63]. It remains an open question how and where information decomposition fits into these and other prior findings, which future work should investigate.

In addition to neurons that respond selectively to particular stimuli (pure selectivity), there also exist, especially in higher-order cortical regions such as the prefrontal cortex and hippocampus, neurons that respond to diverse sets of stimuli and tasks, rather than performing a single specialized function. These sets of neurons are said to have (nonlinear) mixed selectivity (NMS), exhibiting complex responses to different task parameters. Recent work has shown that NMS neurons support flexible behavior and complex cognition [64]. We suggest that synergistic information processing may be closely related to NMS neurons in the brain. In particular, we predict that neurons exhibiting NMS are synergistic and neurons exhibiting pure selectivity have predominantly unique and redundant representations. Future work should study the relation of information decomposition to neuronal selectivity, as it could provide new approaches for understanding the information processing of various neuronal populations.

## Limitations and future work

PID is a relatively recent theoretical framework, and as such its practical applications are often faced with certain limitations [32]. Perhaps the most important of these is that the number of PID atoms grows super-exponentially with the number of sources [6]—a problem that we bypass here by averaging across small subsets of neurons. Furthermore, it seems increasingly clear that there is no universal redundancy function, and that different formulations capture different aspects of multivariate information. In this paper we address these issues by validating our analyses by using two different redundancy functions ($I_{MMI}$ [31], and $I_{min}$ [6]). We are excited by other recent work that has developed PID for application in multivariate settings and continuous variables [18,65–67] and has applied such multivariate measures to deep neural networks [23]. Future work should investigate ways of scaling PID to larger systems and clarify the relationship between different redundancy functions.

In addition to the PID-specific issues above, there is a more general difficulty that arises when computing information-theoretic quantities from data: to estimate these, one needs to know (or accurately approximate) the probability distribution of the observed data [68]. This is particularly challenging in neural networks with non-linearities, such as rectified linear units (ReLUs). In the specific case of neural networks, this issue has caused extensive debate [15,16]. Here we mitigate this problem by verifying that our results are consistent with two estimators, both discrete and continuous, with different hyperparameter settings, and with different discretization approaches, over a range of bins and widths. Nonetheless, future work should elaborate on this direction by either using more sophisticated estimators [68], or using networks where distributions are easier to calculate analytically (e.g., deep linear networks [69,70]).

Finally, it is worth mentioning that all the networks used here are small (on the order of tens of neurons) compared to state-of-the-art networks (typically on the order of many thousands, or more). Thus, future work should investigate to what extent the results presented here generalize to larger networks—as the ones often used nowadays in a range of applications. That being said, the consistency of our results across training regimes (supervised learning, and reinforcement learning with and without recurrent networks), across hidden layer sizes, and across experiment suites (logic gates, Animal-AI, NeuroGym) constitute encouraging preliminary evidence. After the limitations of scaling and approximation above have been overcome, future work should try to replicate these results with larger networks, more complex tasks, and different architectures and training hyperparameters.

In this work we used information decomposition to analyze how artificial neural networks process information in a variety of experimental settings. By studying the learning of logic gates, we found that performance depends on synergistic neurons in tasks requiring the integration of information, and that randomly turning off neurons during training with dropout increases redundancy and robustness while minimizing task-irrelevant features. Using a 3D environment with simulated physics, we showed that synergy is driven by the integration of additional sources of information in a complex reinforcement learning setting. Finally, by studying decision-making tasks inspired by cognitive neuroscience, we found that synergy is specifically increased by the solving of multiple incongruent tasks and the capacity to integrate information from several modalities. Based on these findings, we suggest specific functional roles for PID atoms in the context of learning: redundancy for robustness, unique information for specialization, and synergy for modality integration, flexibility, and efficient encoding. These results lay down foundations to study how learning scenarios modulate information processing modalities, while providing insights into existing cognitive neuroscience results— where synergy is especially high in the most functionally flexible cortical regions (association cortices in frontal and parietal regions), and redundancy has been found in the most functionally specialized and robust areas of the cortex, including sensory and motor regions.

## Materials and methods

### Model architectures

For each task and setting, an ensemble of 10 models is trained, with each network initialized using a different random seed. Additionally, all models use either rectified linear unit (ReLU) or leaky ReLU (for computing continuous measures) activation functions between each linear layer to avoid the compression of mutual information associated with double-saturating nonlinearities, as described in [16]. Network architectures varied slightly between experiments, but in general they consisted of either one or two layers with ten neurons each.

### Quantifying information decomposition

We compute information decomposition over the sampled activations of a network during testing, with its weights frozen (i.e., after first training it on a task). For curriculum tasks, models are tested on all configurations within the curriculum they are trained on, ensuring a fair comparison between different training points within a curriculum. The activations sampled during the testing phase are then used to compute distributions over the activity of the network, used for quantifying redundancy and synergy. Approximating probability distributions is particularly challenging in neural networks with nonlinearities and we thus use two estimators, different hyperparameter settings, and different discretization strategies to ensure consistency in our results. For our experiments performed in NeuroGym, we use a Gaussian copula for the information-theoretic estimation, which is a continuous estimator better-suited for the

large and continuous observation space. We did not use a Gaussian copula for the other experiments (logic gate and Animal-AI experiments; see below) because their discrete observation space is incompatible with this method. Therefore, for these experiments we use a discrete estimator and discretize activations via binning.

Although the discretization and selection of sources-target pairing differ, the methods for information decomposition calculations remain the same across all settings. We refer to a source as being either an individual neuron, a dimension of input, or several dimensions of input grouped as a single random variable. Thus, a set of sources refers to either a set of neurons in a layer, a set of input dimensions, or a set of several dimensions of input that are considered as separate sources. In all cases, the target corresponds to the subsequent layer of neurons considered jointly.

**Discretization of activations.**   Discretization of continuous values, such as the activations of neural networks, presents a challenge for accurately approximating probability distributions, as the choices of bin width and range are likely to influence resulting measures. To address potential issues with binning selection, in addition to the discretization method we employ specific to each experimental setting (logic gate and Animal-AI, specified in the methods section of the corresponding setting), we also compute PID measures using 4 other binning strategies and compare them to summary statistics of our main results (Table A in S1 Text), finding the results to be qualitatively similar.

In particular, for each layer and each model seed, we select the maximum of values to be binned across as the 3rd quartile + 1.5×IQR of the set of sampled activations, where IQR is the interquartile range. The minimum is selected as the maximum between the 1st quartile —1.5×IQR and 0, since our ReLU activations constrain all samples to be nonnegative. We then take an even-frequency split of the samples between the resulting range according to the number of bins (e.g., for 2 bins splitting by the median, for 4 bins splitting by quartiles, etc.). We employ this method across 3, 4, 5, and 10 bins. In the Animal-AI experiments, we discretize the continuous parts of the input with the same number of bins used for the layer discretization based on the IQR, but with evenly spaced bins across the entire observation space.

**Discrete measures.**   For a given set of sources and target, their corresponding discretized activations are used to compute a probability distribution by counting the number of occurrences of each joint sources-target state and using the plug-in estimator [71]. We use the *dit* library [72] to create the distribution and compute the measures of interest.

Although PID proposes the distinction of unique, redundant, and synergistic information, it does not specify a method for computing these measures. Consequently, a number of different formulae have been proposed that capture different aspects of multivariate information, although there is currently no general agreement on a particular measure. Thus, for completeness, we compute all measures using two different redundancy functions: $I_{\min}$ [6] and $I_{\text{MMI}}$ [31]. To provide some intuition, $I_{\text{MMI}}$ computes redundancy as the minimum amount of information any single source provides about the target and synergy as the minimum amount of information lost about the target by removing any single source from the entire set of sources. $I_{\min}$ computes redundancy and synergy similarly, but by instead taking the *expected value* with respect to the target of the minimum amount of information provided by any single source or lost by removing any single source.

We find both measures to be consistent with each other across all experimental settings, with $I_{\text{MMI}}$ yielding slightly higher synergy values. For the purposes of display, we only include $I_{\text{MMI}}$ measures in the body of the text and refer the reader to Figs L-X in S1 Text for all figures replicated using the $I_{\min}$ redundancy function.

For a set of sources $\mathbf{X} = \{X_1, X_2, \ldots, X_M\}$ and a target $Y$ with $N$ possible values, redundancy and synergy for $I_{\text{MMI}}$ are defined as:

$$R_{\text{MMI}}(\mathbf{X}; Y) = \min_i I(X_i; Y) \tag{4}$$

$$S_{\text{MMI}}(\mathbf{X}; Y) = I(\mathbf{X}; Y) - \max_{\mathbf{A} \subseteq \mathbf{X}: |\mathbf{A}| = M-1} I(\mathbf{A}; Y) \tag{5}$$

Similarly, redundancy and synergy for $I_{\min}$ are defined as:

$$R_{\min}(\mathbf{X}; Y) = \sum_{j=1}^{N} p(y_j) \min_i I(X_i; Y = y_j) \tag{6}$$

$$S_{\min}(\mathbf{X}; Y) = I(\mathbf{X}; Y) - I_{\max}(\mathbf{A} \subseteq \mathbf{X} : |\mathbf{A}| = M - 1; Y) \tag{7}$$

Where $I_{\max}$ is defined as:

$$I_{\max}(\mathbf{X}; Y) = \sum_{j=1}^{N} p(y_j) \max_i I(X_i; Y = y_j) \tag{8}$$

**Continuous measures.** To compute PID in the NeuroGym experiments we use the Gaussian Copula Mutual Information (GCMI) estimator by [73], which can deal with some of the nonlinearities introduced by the neurons' activation function. In the Gaussian case, $I_{\text{MMI}}$ and $I_{\min}$ are known to be very similar (in fact proven to be identical in some cases [31]), so for simplicity we run all analyses with $I_{\text{MMI}}$. We compute the average 2nd-order synergy over a random sample of 45 pairs.

**Full- vs 2nd-order decomposition.** One challenge of PID is that the number of PID atoms grows super-exponentially with the number of sources [6]. We bypass this here by averaging across small subsets of sources and using small networks. All calculations are performed over sets of sources of size $K$, where $K$ is either the cardinality of the full set of sources $M$ (either the whole layer or the whole input space; referred to as full-order); or $K = 2$ (2nd-order) (Fig 1C). More specifically, for a set of sources $\mathbf{X} = \{X_1, X_2, \ldots, X_M\}$ and target $Y$, the $K$-order redundancy $R^{(K)}$ is defined as the average of the redundancy of subsets of sources of cardinality $K$:

$$R^{(K)}(\mathbf{X}; Y) = \langle R(\mathbf{A}; Y) \rangle_A, \text{ with } \mathbf{A} \subseteq \mathbf{X}, |\mathbf{A}| = K$$

In the case of 2nd-order, $K = 2$ and in full-order, $K = M$ (therefore there is only one set being considered in the full-order case). In other words, for the case of full-order decomposition, there is only one set containing all source variables and the decomposition is taken over the joint mutual information of all source variables. For 2nd-order measures, all calculations are performed between pairs of sources (i.e., over subsets of the source set—a hidden layer or input—of cardinality 2). Thus, a 2nd-order value is computed using only 2 elements of a set as sources, rather than the full set. Performing this operation over all possible combinations of pairs and computing the mean gives the average 2nd-order measure. When the system grows too large to efficiently compute all possible combinations, this value can additionally be approximated by uniformly sampling pairs of combinations.

We show that full-order and 2nd-order measures exhibit similar qualitative behavior in response to dropout and task (Figs A and B in S1 Text). However, both our results and prior work [25] suggest that redundancy and synergy are more prevalent at smaller orders, especially for small networks. By computing average 2nd-order synergy, we can partially capture how

synergistically-biased [74] a set of sources is—with higher $2^{nd}$-order synergy, the PID lattice will have more synergistically-interacting atoms than redundantly-interacting atoms. Given these properties, we use $2^{nd}$-order measures in the remainder of our experiments.

All of the PID measures shown in the text are $2^{nd}$-order and are normalized by mutual information. Thus, we are specifically showing the proportion of mutual information occupied by each measure—an increase in (normalized) synergy is in favor of either redundancy or unique information, which must in turn be reduced.

## Layerwise PID analyses

We perform layerwise comparisons of information decomposition in both the logic gate and Animal-AI experiments (Figs H-K in S1 Text). In logic gate networks, we observe that in both COPY and XOR tasks, the application of dropout increases redundancy with each subsequent layer of the network and that synergy is highest from the input to the first layer. In Animal-AI, we again observe that synergy is highest from the input to the first layer.

We note that these results are preliminary and that the study of network information decomposition at different layer depths requires further work. It is possible that the effects observed are influenced by the size and dimensionality of input and action space, and by the size of each layer (e.g., sampling two sources out of two possible neurons versus ten possible neurons), which is not controlled for.

## Logic gate experiments

The data used for the COPY and XOR logic gate experiments are generated as a two-dimensional binary input with a binary output. The label of each COPY gate input corresponds to the copying of the first input and the label of the XOR gate input corresponds to the parity of both inputs.

Our models are small feedforward networks, with two layers consisting of ten neurons each. Dropout is only applied during initial training and not during testing or lesioning evaluation. Each model is trained to convergence. We subsequently test and compute various information decomposition measures.

Each activation sampled during testing is discretized using 3 bins in the range of [0,5]. We use 3 bins to ensure a sufficient number of samples in each source-target pair. The range of bins is chosen based on empirical observations of the network activations being heavily concentrated within this range.

For the lesioning experiments, we additionally compute the average pairwise synergy for each neuron. For a particular neuron, this is performed by computing all $2^{nd}$-order synergy values that include the neuron as one of the sources and calculating the mean.

## Animal-AI experiments

The experiments conducted in the Animal-AI Environment [37] are performed with proximal policy optimization (PPO) models [75] using *Stable-Baselines3* [76]. The actor-critic networks of the models consist of two feedforward layers with ten neurons in each layer, identical to those used in our logic gate experiments.

During training, we evaluate and compute synergy for each model at each task threshold. For all tasks, the threshold is chosen as the point at which the model successfully reached the maximum reward or the maximum number of steps per task (2 million steps).

We constrain the observation space to three object-oriented raycasts, each being a one-hot vector indicating the type of object hit by the raycast and its distance normalized by the size of the arena, and an additional vector relaying information about the agent's health, velocity, and

global position. The raycasts are projected 90 degrees apart—directly in front of the agent, to its left, and to its right. Thus, the agent has full access to the information required to solve the task at the first time step of the episode, preventing the addition of bias that could be introduced by input bits being occluded and the need for integration of input information over time. The raycast observation space occludes the contents of both pits (positive/negative reward), which would otherwise be visible to an agent receiving the full pixel observation space. It also allows for the use of small feedforward networks, rather than larger convolutional layers necessitated by a full pixel input. The small observation space also facilitates our models' (with small network parameter spaces) ability to solve the given tasks in a complex three-dimensional environment.

We use the same basic design shown in Fig 3 for all tasks created. In the pit corresponding to the correct output lies an occluded object with a reward of 4 and in the other pit lies an occluded 'death zone' with a reward of -1, both of which terminate the episode upon being reached. The agent's movement is constrained to the platform and one pit and therefore successful completion of the task is contingent on using the information relayed by the bit-representing barriers.

Using this design, we modify the placement of the positive and negative rewards according to the logic gate task being performed. Agents are placed on a short platform to restrict their possible state space and simplify the task, serving as a minimal baseline for solving logic gates in a RL setting. The 3-Bit XOR task explored the effect of integrating an additional source by placing a third input-source barrier in front of the agent (Fig 3C). Finally, we create three tasks (Distance XOR-10, 20, and 30) by using the basic task design and elongating the platform to lengths of 10, 20, and 30 arena units, increasing the distance between the agent and the reward as a method of adding more difficulty to the logic gate task without the addition of sources (Fig 3D). Our curriculum tasks consist of the combination of 2-Bit XOR to 3-Bit XOR; and 2-Bit XOR to Distance 10, 20, and 30 XOR.

Synergy is computed in the actor network of the PPO model. Because the observation space exceeds 20 dimensions, synergy cannot be efficiently computed over the entire input using our measures. However, due to the modularity of the raycast input, grouping dimensions based on object-related information is likely to yield more interpretable measurements. Thus, to compute synergy from the input sources to the first linear layer target, we treat each raycast as a single source with the vector dimension for normalized distance being discretized with 3 bins from [0,1]. Additionally, the global position is also treated as a single source and discretized using 5 bins from [0,40] (40 being the length of the arena). We compute the average $2^{nd}$-order synergy between all combinations of source pairs of raycasts (shown in the main text) and source pairs of each raycast and the global position (Fig G in S1 Text), which yield similar values. The average $2^{nd}$-order synergy is then computed for the rest of the network.

## NeuroGym experiments

Our RNN models consist of a single recurrent unit with a hidden layer size of ten neurons using a leaky ReLU activation. During training, decision-making actions are weighted by a favor of 20 in the cross-entropy loss compared to the action corresponding to fixation, due to their relative scarcity in the training process. The observation space is modified to include a binary indicator signifying the task being performed. We test and compute synergy after training for 80,000 total steps (40,000 per task in the sequential protocol; 80,000 total in the interleaved protocol). Each task is trained using supervised learning and we modify tasks with variable episode timing to be of fixed length.

Unlike our other experiments, the tasks in NeuroGym are stochastic and have a much larger task-space. To ensure sufficient activation sampling, we test models on a total of 100 trials. We use both the input and hidden layer at each time step as sources and the hidden layer at the following time step as the target.

### Statistical analysis

We perform independent samples t-tests when comparing different models and paired samples t-test when comparing the same models at different points during training. Additionally, we perform a Benjamini-Hochberg False Discovery Rate correction to account for multiple comparisons made in our lesion experiments and when comparing interleaved and sequential protocols across pairs of tasks in NeuroGym.

### Supporting information

**S1 Text. Fig A in S1 Text. Full-order and 2$^{nd}$-order measures exhibit similar behavior for the COPY task.** Values represent individual data points, and means ± SEM. **Fig B in S1 Text. Full-order and 2$^{nd}$-order measures exhibit similar behavior for the XOR task.** Values represent individual data points, and means ± SEM. **Fig C in S1 Text. Networks with different layer sizes (ten versus twenty neurons) exhibit similar behavior for the COPY task (2$^{nd}$-order).** Values represent individual data points, and means ± SEM. **Fig D in S1 Text. Networks with different layer sizes (ten versus twenty neurons) exhibit similar behavior for the XOR task (2$^{nd}$-order).** Values represent individual data points, and means ± SEM. **Fig E in S1 Text. Effects of lesions and dropout on network information profiles replicate in larger networks (twenty neurons rather than ten).** (a) (****$P<0.0001$, independent samples $t$ test; $n = 20$). Values represent probability density functions. (b) (*$P<0.05$, **$P<0.01$, ***$P<0.001$, ****$P<0.0001$, paired samples $t$ test with Benjamini-Hochberg False Discovery rate correction; $n = 20$). Values represent means ± SEM. **Fig F in S1 Text. Dropout removes irrelevant redundant and synergistic information about the input in the COPY task, but not the XOR task (*$P<0.05$, **$P<0.01$, independent samples $t$ test; $n = 20$).** Values represent probability density functions. **Fig G in S1 Text. Relation of compositional tasks and synergy in Animal-AI using pairwise raycast-position sources.** (a) (**$P<0.01$, paired samples $t$ test; $n = 20$). Values represent probability density functions. (b) (n.s., not significant, paired samples $t$ test; $n = 20$). Values represent probability density functions. (c) Values represent probability density functions. **Fig H in S1 Text. Layer-wise comparison for COPY task with different levels of dropout applied (*$P<0.05$, **$P<0.01$, ***$P<0.001$, ****$P<0.0001$, independent samples $t$ test; $n = 20$).** Values represent probability density functions. **Fig I in S1 Text. Layer-wise comparison for XOR task with different levels of dropout applied (*$P<0.05$, **$P<0.01$, ****$P<0.0001$, independent samples $t$ test; $n = 20$).** Values represent probability density functions. **Fig J in S1 Text. Layer-wise comparison at the end of training for Animal-AI tasks using pairwise raycast sources (**$P<0.01$, ****$P<0.0001$, independent samples $t$ test; $n = 20$).** Values represent probability density functions. **Fig K in S1 Text. Layer-wise comparison at the end of training for Animal-AI tasks using pairwise raycast-position sources (**$P<0.01$, ****$P<0.0001$, independent samples $t$ test; $n = 20$).** Values represent probability density functions. **Fig L in S1 Text. Full-order and 2$^{nd}$-order measures exhibit similar behavior for the COPY task replicated using $I_{min}$ redundancy function.** Values represent individual data points, and means ± SEM. **Fig M in S1 Text. Full-order and 2nd-order measures exhibit similar behavior for the XOR task replicated using $I_{min}$ redundancy function.** Values represent individual data points, and means ± SEM. **Fig N in S1 Text. Networks with different layer sizes (ten versus twenty neurons) exhibit similar behavior for the**

**COPY task (2<sup>nd</sup>-order) replicated using $I_{min}$ redundancy function.** Values represent individual data points, and means ± SEM. **Fig O in S1 Text. Networks with different layer sizes (ten versus twenty neurons) exhibit similar behavior for the XOR task (2<sup>nd</sup>-order) replicated using $I_{min}$ redundancy function.** Values represent individual data points, and means ± SEM. **Fig P in S1 Text. Dropout removes irrelevant redundant and synergistic information about the input in the COPY task, but not the XOR task (*$P<0.05$, **$P<0.01$, independent samples *t* test; *n* = 20) replicated using $I_{min}$ redundancy function.** Values represent probability density functions. **Fig Q in S1 Text. Effects of lesions and dropout on network information profiles replicated using $I_{min}$ redundancy function.** (a) (*$P<0.05$, ***$P<0.001$, ****$P<0.0001$, independent samples *t* test; *n* = 20). Values represent probability density functions. (b) (*$P<0.05$, **$P<0.01$, paired samples *t* test with Benjamini-Hochberg False Discovery rate correction; *n* = 20). Values represent means ± SEM. **Fig R in S1 Text. Effects of lesions and dropout on network information profiles replicate in larger networks (twenty neurons rather than ten) replicated using $I_{min}$ redundancy function.** (a) (****$P<0.0001$, independent samples *t* test; *n* = 20). Values represent probability density functions. (b) (*$P<0.05$, **$P<0.01$, ***$P<0.001$, ****$P<0.0001$, paired samples *t* test with Benjamini-Hochberg False Discovery rate correction; *n* = 20). Values represent means ± SEM. **Fig S in S1 Text. Relation of compositional tasks and synergy in Animal-AI using pairwise raycast-position sources replicated using $I_{min}$ redundancy function.** (a) (**$P<0.01$, paired samples *t* test; *n* = 20). Values represent probability density functions. (b) (n.s., not significant, paired samples *t* test; *n* = 20). Values represent probability density functions. (c) Distance XOR refers to Distance 10 XOR. Values represent probability density functions. **Fig T in S1 Text. Relation of compositional tasks and synergy in Animal-AI using pairwise raycast-position sources replicated using $I_{min}$ redundancy function.** (a) (**$P<0.01$, paired samples *t* test; *n* = 20). Values represent probability density functions. (b) (n.s., not significant, paired samples *t* test; *n* = 20). Values represent probability density functions. (c) Distance XOR refers to Distance 10 XOR. Values represent probability density functions. **Fig U in S1 Text. Layer-wise comparison for COPY task with different levels of dropout applied replicated using $I_{min}$ redundancy function (**$P<0.01$, ***$P<0.001$, ****$P<0.0001$, independent samples *t* test; *n* = 20).** Values represent probability density functions. **Fig V in S1 Text. Layer-wise comparison for XOR task with different levels of dropout applied replicated using $I_{min}$ redundancy function (****$P<0.0001$, independent samples *t* test; *n* = 20).** Values represent probability density functions. **Fig W in S1 Text. Layer-wise comparison at the end of training for Animal-AI tasks using pairwise raycast sources replicated using $I_{min}$ redundancy function (*$P<0.05$, ****$P<0.0001$, independent samples *t* test; *n* = 20).** Values represent probability density functions. **Fig X in S1 Text. Layer-wise comparison at the end of training for Animal-AI tasks using pairwise raycast-position sources replicated using $I_{min}$ redundancy function (*$P<0.05$, ***$P<0.001$, ****$P<0.0001$, independent samples *t* test; *n* = 20).** Values represent probability density functions. **Table A in S1 Text. Summary statistics for logic gate and Animal AI experiments across different binning strategies for discretization (*$P<0.05$, **$P<0.01$, ***$P<0.001$, ****$P<0.0001$, n.s., not significant).** P-values are not corrected for multiple comparisons.
(DOCX)

## Acknowledgments

We would like to thank Pedro Urbina Rodriguez for pointing us to useful references during our revision of the manuscript.

## Author Contributions

**Conceptualization:** Matthew Crosby, Pedro A. M. Mediano.

**Formal analysis:** Alexandra M. Proca.

**Investigation:** Alexandra M. Proca.

**Methodology:** Alexandra M. Proca.

**Software:** Alexandra M. Proca.

**Supervision:** Matthew Crosby, Pedro A. M. Mediano.

**Visualization:** Alexandra M. Proca.

**Writing – original draft:** Alexandra M. Proca.

**Writing – review & editing:** Alexandra M. Proca, Fernando E. Rosas, Andrea I. Luppi, Daniel Bor, Matthew Crosby, Pedro A. M. Mediano.

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
