## [Decision Letter · Decision Letter 0]

28 Dec 2023

Dear Miss Proca,

Thank you very much for submitting your manuscript "Synergistic information supports modality integration and flexible learning in neural networks solving multiple tasks" for consideration at PLOS Computational Biology.

As with all papers reviewed by the journal, your manuscript was reviewed by members of the editorial board and by several independent reviewers. In light of the reviews (below this email), we would like to invite the resubmission of a significantly-revised version that takes into account the reviewers' comments.

We cannot make any decision about publication until we have seen the revised manuscript and your response to the reviewers' comments. Your revised manuscript is also likely to be sent to reviewers for further evaluation.

Sincerely,

Marcus Kaiser, Ph.D.

Academic Editor

PLOS Computational Biology

Thomas Serre

Section Editor

PLOS Computational Biology

Reviewer's Responses to Questions

**Comments to the Authors:**

Reviewer #1: The review uploaded as attachment.

Reviewer #2: Proca et al. employ a recent extension of information theory, called partial information decomposition to the analysis of distributed computation in neural systems learning multiple tasks. They show that the mutual information between subsequent processing stages (or inputs and processing stages) moves into PID-atoms of higher synergy when networks learn multiple diverse tasks. They conclude that synergy in these networks enables flexible (and efficient) learning. While the efficiency gain via synergistic modes of operation may have been expected based on purely theoretical considerations about the size of possible coding spaces, the flexibility is indeed a novel and important result.

This is a timely and important study that highlights the shortcomings of classic information quantities in the analysis of distributed computation and shows how to overcomes these shortcomings via partial information decomposition. The example cases presented are well chosen and of high relevance, and the simulations and analysis are preformed thoroughly. As such I think this manuscript presents an important advance in the field. I would like to stress this aspect here, as I will list a range of criticisms below that need to be addressed, and may sound harsh, but which also can be addressed and should in my opinion not stand in the the way of the ultimate success of this study.

Since it is one of the first studies of its kind, and therefore lays a foundation for future investigations of this kind, is has to be held to the higher standards than usual in order to avoid that the field of PID analyses of distributed computation drifts into some sort of ad-hoc, hand waving ‘reading of tea-leaves’ as has for example happened with the now infamous cross-frequency coupling analyses of the past decade.

To this end I think it is mandatory that the following issues be thoroughly addressed:

1. I think it is not OK to totally gloss over the problem of various PID measures in the Background section on partial information decomposition, and to defer any further discussion to the appendix/supplementary. This is because to date >25 different PID measures exist, with many of them shown to solve highly specific PID problems. The problem solved by a particular PID is tightly related to what is called the operational interpretation of the respective PID, i.e. its very meaning. For example, the popular measures I_BROJA and I_CCS have thorough decision or game theoretic operational interpretations, respectively, but using I_BROJA in a game theoretic setting would be just as wrong as using I_CCS in the decision theoretic setting, as has been demonstrated explicitely. Thus, the application of a specific PID measure to a neuroscience or machine learning question has to be justified thoroughly by the corresponding operational interpretation of the measure.

It is also well known that various PID measures can strongly disagree in certain problem settings [1], and that this must be this way, simply because their measurements have entirely different interpretations. So I would like to ask the authors to make explicit the link between the operational interpretation of the two PID measures they used in their study and the problem setting addressed in the manuscript.

2. The issue of binning and the problems related to binning choices should be treated better. It is obvious from the simple observation that in the limit of very few bins PIDs tend to turn out too high levels of redundancy, while in the limit of too many bins and estimation from finite data will be almost certainly a one-to-one mapping of each specific pair of inputs to one specific output for that pair, and thus, just synergy. This seems to prohibit the use of binned variables in a PID setting – unless the binning can be motivated extrinsically, i.e. in the sense that the computational system itself uses that binning or has a sensitivity reflecting that binning. While I accept the difficulty that so far only very few PID measures and estimators for systems of mixed discrete and continuous variables exist (e.g. [4,5], but there is no code available at present), this problem cannot simply be ignored. A more acceptable way to tackle the problem of mixed discrete-continuous observation spaces would perhaps be to use a small amount of Gaussian noise on the discrete parts of the observation space and the Gaussian copula method instead of some ad-hoc binning; a less desirable alterative approach would be to present results over a wide range of bin numbers or widths and to thoroughly discuss the binning issue. I think this point is very important so as not to set a bad precedent that is then in the future cited in order to justify the practice of binning in the PID setting, as this would certainly damage the field.

Minor suggestions for improvement

3. The references to more advanced measures seems to be lacking important recent developments, both, for multivariate PIDs (e.g. [2]), PIDs for continuous variables (e.g. [3]) and the combination of these properties (e.g. [4,5]) – just to name a few. In this context it may also be in order to mention that some of these multivariate measures indeed have been applied to deep neural networks – e.g. up to five-input PIDs [7].

4. A pointer to the overall structure of the field of PID measures would be in order, to give the reader a feeling for the possibilities and limitations of the concept. The introductory review [6] seems to be a frequently cited reference to consider here but any other recent review on the overall structure of the field would be just as fine.

5. The use of PID has rapidly gained momentum in the machine learning world. It may be good to stress the importance of the method applied in the present study and thus the validity of the presented approach to highlight this general interest and acceptance of PID in the machine learning world by including some references documenting this. One of the reasons for why this may be beneficial for the readers of the current manuscript may be that they can then find the approach that best fits their particular use case in terms of operational interpretations – as explained above.

6. Unfortunately it remains unclear to me how the full order PID decomposition was performed. The authors write: “ All calculations are performed over sets of sources of size K, where K is either the cardinality of the full set of sources (either the whole layer or the whole input space; referred to as full-order); or K = 2 (2nd-order)…” When K is the cardinality of the full set of sources, then there is only one such set (containing all source variables), and we thus seem to obtain a joint mutual information – is this what was done? If not please rephrase and include a formula for the full-order decomposition, and the maximum number of atoms resulting from the cardinality used in the experiments. To restate my problem: I do understand the 2nd-order decomposition, but not the full-order counterpart.

References:

[1] Kay, J. W., Schulz, J. M., & Phillips, W. A. (2022). A comparison of partial information decompositions using data from real and simulated layer 5b pyramidal cells. Entropy, 24(8), 1021.

[2] Chicharro, D., & Panzeri, S. (2017). Synergy and redundancy in dual decompositions of mutual information gain and information loss. Entropy, 19(2), 71.

[3] Pakman, A., Nejatbakhsh, A., Gilboa, D., Makkeh, A., Mazzucato, L., Wibral, M., & Schneidman, E. (2021). Estimating the unique information of continuous variables. Advances in neural information processing systems, 34, 20295-20307.

[4] Schick-Poland, K., Makkeh, A., Gutknecht, A. J., Wollstadt, P., Sturm, A., & Wibral, M. (2021). A partial information decomposition for discrete and continuous variables. arXiv preprint arXiv:2106.12393.

[5] Ehrlich, D. A., Schick-Poland, K., Makkeh, A., Lanfermann, F., Wollstadt, P., & Wibral, M. (2023). Partial Information Decomposition for Continuous Variables based on Shared Exclusions: Analytical Formulation and Estimation. arXiv preprint arXiv:2311.06373.

[6] Gutknecht, A. J., Wibral, M., & Makkeh, A. (2021). Bits and pieces: Understanding information decomposition from part-whole relationships and formal logic. Proceedings of the Royal Society A, 477(2251), 20210110.

[7] Ehrlich, D. A., Schneider, A. C., Priesemann, V., Wibral, M., & Makkeh, A. (2023). A measure of the complexity of neural representations based on partial information decomposition. Transactions on Machine Learning Research, 5.

Reviewer #3: The paper presents a really nice analysis of information encoding in neural network models that learn tasks. The main result is the emergence of more and more synergistic neurons as the networks learn to perform more and more refined tasks.

The analyses are well done and the conclusions well-reasoned and well-taken.

I only have few requests for clarification.

The authors cite extensively their own work and that of others about synergy in brain data. However, most of the cited literature is describing analyses of aggregate neural signals, such as fMRI that have relatively little to reveal about computations at the population or single neuron levels. Surprisingly, the authors do not consider instead the coding of information with single neuron resolution, which been studied as well (e.g. Sherrill et al (2021) PLoS Comput Biol 17(7):e1009196. https:// doi. org/ 10. 1371/ journ al. pcbi. 10091 96; Valente et al (2021) Nat Neurosci 24(7):975–986. https:// doi. org/ 10. 1038/ s41593- 021- 00845-1;

Koçillari et al. Brain Informatics (2023) 10:34 https://doi.org/10.1186/s40708-023-00212-9) These single neurons resolution studies would be more relevant and directly comparable to the one presented here in the analysis of artificial neural networks, and some of hese single neuron papers use exactly the same methods (information decompositions) used by the authors for the analysis of neural networks. Importantly, some of these papers have found synergy and recognized its important function. However, all the single neuron papers have also found redundancy and reported convincing evidence of its utility at the neural network computation level. The authors should make more contact with the synergy/redundancy single neuron literature and also discuss more in Discussion te relative merits of synergy and redundancy for neural computations.

In lines 62, the authors write “Synergy and related quantities, such as integrated information (8),”. I would disagree with this statement. Integrated information cannot be reduced to synergy. There are analogies and differences, but I believe that the statement as written is not correct. I invite the authors to rewrite it in a more precise way.

**Have the authors made all data and (if applicable) computational code underlying the findings in their manuscript fully available?**

Reviewer #1: Yes

Reviewer #2: Yes

Reviewer #3: Yes

PLOS authors have the option to publish the peer review history of their article (what does this mean?). If published, this will include your full peer review and any attached files.

Reviewer #1: **Yes: **Timon Kunze

Reviewer #2: No

Reviewer #3: No

Figure Files:

Data Requirements:

Please note that, as a condition of publication, PLOS' data policy requires that you make available all data used to draw the conclusio

---

## [Decision Letter · Decision Letter 1]

18 May 2024

Dear Miss Proca,

We are pleased to inform you that your manuscript 'Synergistic information supports modality integration and flexible learning in neural networks solving multiple tasks' has been provisionally accepted for publication in PLOS Computational Biology.

Best regards,

Marcus Kaiser, Ph.D.

Academic Editor

PLOS Computational Biology

Thomas Serre

Section Editor

PLOS Computational Biology

Reviewer's Responses to Questions

**Comments to the Authors:**

Reviewer #2: This is just a comment on the authors' replies with regard to binning.

While I can accept the answers given in the reply to my concerns about binning in general, I would like to point aut that the follwoing statement: "binned variables in PID. On one extreme, it is clear that there are binary systems without

redundancy (e.g. the XOR gates), so even systems with very few bins can have no redundancy."

Is potentially not relevant to the criticism in question. This is because in a binary XOR gate the number of bins reflects the system's ground truth. Thus, no detrimental effects of binning are expected. My concern related to the fact that in a system with continuous input variables spurious correlations between sources may arise.

This said, I am fully OK with the approach of showing the stability of results over a reasonable range of bins.

Reviewer #3: The authors addressed the concerns well.

**Have the authors made all data and (if applicable) computational code underlying the findings in their manuscript fully available?**

Reviewer #2: Yes

Reviewer #3: Yes

PLOS authors have the option to publish the peer review history of their article (what does this mean?). If published, this will include your full peer review and any attached files.

Reviewer #2: No

Reviewer #3: No

---

## [Editor Report · Acceptance letter]

26 May 2024

PCOMPBIOL-D-23-01804R1 

Synergistic information supports modality integration and flexible learning in neural networks solving multiple tasks

Dear Dr Proca,

I am pleased to inform you that your manuscript has been formally accepted for publication in PLOS Computational Biology. Your manuscript is now with our production department and you will be notified of the publication date in due course.

With kind regards,

Zsofia Freund
